# WHAT ARE EFFECTIVE LABELS FOR AUGMENTED DATA? IMPROVING ROBUSTNESS WITH AUTOLABEL

## ABSTRACT

A wide breadth of research has devised data augmentation approaches that can improve both accuracy and generalization performance for neural networks. However, augmented data can end up being far from the clean data and what is the appropriate label is less clear. Despite this, most existing work simply reuses the original label from the clean data, and the choice of label accompanying the augmented data is relatively less explored. In this paper, we propose `AutoLabel` to automatically learn the labels for augmented data, based on the *distance* between the clean distribution and augmented distribution. `AutoLabel` is built on label smoothing and is guided by the *calibration*-performance over a hold-out validation set. We show that `AutoLabel` is a generic framework that can be easily applied to existing data augmentation methods, including AugMix, mixup, and adversarial training. Experiments on CIFAR-10, CIFAR-100 and ImageNet show that `AutoLabel` can improve models' accuracy and calibration performance, especially under distributional shift. Additionally, we demonstrate that `AutoLabel` can help adversarial training by bridging the gap between clean accuracy and adversarial robustness.

## 1 INTRODUCTION

Deep neural networks are increasingly being used in high-stakes applications such as healthcare and autonomous driving. For safe deployment, we not only want models to be accurate on expected test cases (independent and identically distributed samples), but we also want models to be robust to distribution shift (Amodei et al., 2016) and to not be vulnerable to adversarial attacks (Goodfellow et al., 2014; Carlini & Wagner, 2017; Madry et al., 2017; Qin et al., 2020b). Recent work has shown that the accuracy of state-of-the-art models drops significantly when tested on corrupted data (Hendrycks & Dietterich, 2019). Furthermore, these models are not just wrong on these unexpected examples, but also overconfident – Ovadia et al. (2019) showed that calibration of models degrades under shift. Calibration measures the gap between a model's own estimate of correctness (a.k.a. confidence) versus the empirical accuracy, which measures the actual probability of correctness. Building models that are accurate *and* robust, i.e. can be trusted under unexpected inputs from both distributional shift and adversarial attacks, is a challenging but important research problem.

Improving both calibration under distribution shift and adversarial robustness has been the focus of numerous research directions. While there are many approaches to addressing these problems, one of the fundamental building blocks is data augmentation: generating synthetic examples, typically by modifying existing training examples, that provide additional training data outside the empirical training distribution. A wide breadth of literature has explored what are effective ways to modify training examples, such as making use of domain knowledge through label-preserving transformations (Hendrycks et al., 2020) or adding adversarially generated, imperceptible noise (Madry et al., 2017; Zhang et al., 2019). Approaches like these have been shown to improve the robustness and calibration of overparametrized neural networks as they alleviate the issue of neural networks overfitting to spurious features that do not generalize beyond the i.i.d. test set.

In the broad amount of research on data augmentation, most of it attempts to apply transformations that do not change the true label such that the label of the original example can also be assumed to be the label of the transformed example, without expensive manual review. While there has been a significant amount of work in how to construct such pseudo-examples in input space, there has been relatively little attention on whether this assumption of label-preservation holds in practice and what label should be assigned to such augmented inputs. For instance, many popular methods assign

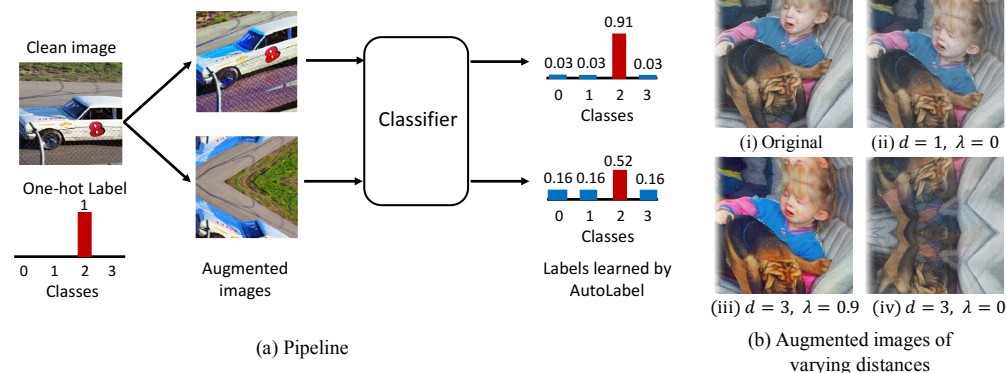

(a) Pipeline

(b) Augmented images of varying distances

Figure 1: (a): An example showing `AutoLabel` assigning different labels to augmented images (e.g., by AugMix (Hendrycks et al., 2020)) based on their transformation distances to the clean image. The label for the true class is automatically learned based on the calibration performance on validation set. (b): Examples of images augmented by AugMix with different distances to the original image.

one-hot targets to both training data as well as augmented inputs that can be quite far away from the training data where even human raters may not be 100% sure of the label. This runs the risk of adding noise to the training process and degrading accuracy.

With this observation, in this paper we investigate the choice of target labels for augmented inputs and propose `AutoLabel`, a method that automatically adapts the confidence assigned to augmented labels, assigning high confidence to inputs close to the training data and lowering the confidence as we move farther away from the training data. Figure 1 (left) gives a high-level overview of our proposed `AutoLabel` along with examples of augmented images of varying distances generating by AugMix (Hendrycks et al., 2020) on the right. Our key contributions are as follows:

- We propose `AutoLabel`, a *distance-based* approach where the training labels are smoothed to different extents based on the distance between the augmented data and the clean data.

- We show that `AutoLabel` is complementary to methods which focus on generating augmented inputs by combining it with popular methods for data augmentation, such as AugMix (Hendrycks et al., 2020), mixup (Zhang et al., 2018) as well as adversarial training (Madry et al., 2017).

- We show that `AutoLabel` significantly improves the calibration of models (and accuracy, although less dramatically) on both clean and corrupted data for CIFAR10, CIFAR100 and ImageNet. In addition, `AutoLabel` also helps bridge the gap between accuracy and adversarial robustness.

## 2 RELATED WORK

**Data Augmentation.**   Recent work has shown that introducing additional training examples can further improve a model's accuracy and generalization (Devries & Taylor, 2017; Cubuk et al., 2018; Yun et al., 2019; Takahashi et al., 2019; Lopes et al., 2019; Zhong et al., 2020). For example, AugMix (Hendrycks et al., 2020) utilizes stochasticity and diverse augmentations, together with a consistency loss over the augmentations, to achieve state-of-the-art corruption robustness. Mixup (Zhang et al., 2018), on the other hand, trains a neural network over convex combinations of pairs of examples and shows improved generalization of neural networks. Furthermore, adversarial training (Goodfellow et al., 2014; Madry et al., 2017; Zhang et al., 2019) can also be thought as a special data augmentation technique aiming for improving model's adversarial robustness. In this paper, we investigate the choice of the target labels for augmented inputs and show how to apply `AutoLabel` to these existing data augmentation techniques to further improve model's robustness.

**Calibration and Uncertainty Estimates.**   A variety of methods have been developed for improving a model's calibration, e.g., post-hoc calibration by temperature scaling (Guo et al., 2017) and multi-class Dirichlet calibration (Kull et al., 2019). Model's predictive uncertainty can also be quantified using Bayesian neural networks and approximate Bayesian approaches, e.g., variational inference (Graves, 2011; Blundell et al., 2015), MCMC sampling based on stochastic gradients (Welling & Teh, 2011), and dropout-based variational inference (Kingma et al., 2015; Gal & Ghahramani, 2016). In addition to calibration over in-distribution data, more recently, Ovadia et al. (2019) show that model calibration can further degrade under unseen data shifts, where ensemble of deep neural networks (Lakshminarayanan et al., 2017) is shown to be most robust to dataset shift.

On the other hand, several data augmentation methods have also been shown to improve model's calibration under data shifts. For example, AugMix is shown to improve uncertainty measures on corrupted image classification benchmarks (Hendrycks et al., 2020). Thulasidasan et al. (2019) demonstrate that neural networks trained with mixup are significantly better calibrated under dataset shift, and are less prone to over-confident predictions on out-of-distribution data.

**Label Smoothing.** Label smoothing, initially proposed in Szegedy et al. (2016), is used to prevent a model from being too over-confident in its predictions, thus improving its generalization ability. It has been shown by Müller et al. (2019); Thulasidasan et al. (2019) that label smoothing can also effectively improve the quality of a model's uncertainty estimates. Our work is most closely related to the adaptive label smoothing algorithm in (Qin et al., 2020c). Qin et al. (2020c) observe the connection between adversarial robustness and uncertainty, and propose an algorithm for adaptively updating the amount of label smoothing based on the adversarial vulnerability of *clean data* to improve model's calibration. In contrast, we propose to adaptively smooth the labels for *augmented data* based instead on the distance to the clean training data, and show it can further improve a model's accuracy, calibration and adversarial robustness.

## 3 NOTATION AND CALIBRATION METRIC

**Notations:** Given a clean dataset $\mathcal{D} = \{(x_i, y_i)\}_{i=1,\cdots,m}$, we consider a classifier $f(\cdot)$ for a $K$-class classification problem, where $y \in \{1, ..., K\}$. The one-hot encoding of the label is denoted as $\hat{y} \in \{0, 1\}^K$, where the label for the true class $\hat{y}_{k=y} = 1$ and $\hat{y}_{k \neq y} = 0$ for others. Let $f_k(x)$ denote the predicted probability for the $k$-th class. We use $f(x) := \text{argmax}_k f_k(x)$ to represent the predicted class and $c(x) := \max_k f_k(x)$ as model's confidence of the predicted class. In addition to the training data $\mathcal{D}$, we also have a clean validation set $\mathcal{D}_V$ drawn i.i.d. from the same distribution.

**Expected Calibration Error (ECE)** We use Expected Calibration Error (ECE) to measure a model's calibration performance as (Guo et al., 2017; Ovadia et al., 2019). This metric measures how well aligned the average accuracy and the average predicted confidence are, after dividing the input data into $R$ buckets: $\text{ECE} = \sum_r^R \frac{|B_r|}{m} |\text{acc}(B_r) - \text{conf}(B_r)|$, where $B_r$ indexes the $r$-th confidence bucket and $m$ denotes the data size, the accuracy and the confidence of $B_r$ are defined as $\text{acc}(B_r) = \frac{1}{|B_r|} \sum_{i \in B_r} \mathbf{1}(f(x_i) = y_i)$ and $\text{conf}(B_r) = \frac{1}{|B_r|} \sum_{i \in B_r} c(x_i)$.

## 4 ALGORITHM

We now dive into our algorithm for more effectively setting labels for augmented data. We will first discuss the general `AutoLabel` algorithm and then discuss how it can be applied to three different data augmentation techniques.

### 4.1 AUTOLABEL

Given a data augmentation technique `Aug` that takes in an example $x$ and outputs a transformed version, `AutoLabel` is mainly composed of two components: (1) a measure of the transformation distance from the original input example, and (2) a subroutine for updating labels of the augmented examples during training. Many data augmentation approaches have hyperparameters that reflect how large the transformation should be; we refer to this as the transformation distance $s$. As examples, which we will discuss in-depth below, this can take the form of the number of transformations in AugMix (Hendrycks et al., 2020) or the norm of the adversarial perturbation in Madry et al. (2017). Put together, we use `Aug`$(x, s)$ to denote the generation of a transformed example.

*Our key insight is that the confidence in the labels associated with the augmented data likely depends on how significant the transformation is.* To make use of this insight, we would like to assign different labels for the augmented training data based on their transformation distances. The labels can be updated according to the calibration of the model on the augmented validation data. To this end, we must be able to split both the training data and validation data into buckets for measuring calibration. Here, we will discretize the transformation distance $s$ into $N$ buckets $\{S_1, \cdots, S_N\}$ where each $S_n$ is a range, and we can generate augmented validation data for bucket $S_n$ by sampling a distance uniformly in that range $s \sim \mathcal{U}(S_n)$ to generate `Aug`$(x, s)$.

`AutoLabel` takes a label smoothing approach where the amount of smoothing depends on the calibration against a similarly augmented validation set $\mathcal{Q}(S_n) = \{(\texttt{Aug}(x_i, s), y_i) | (x_i, y_i) \in$

Table 1: Distance measure under various data augmentation methods used by `AutoLabel`.

| Augmentation Method | AugMix | mixup | Adv. Training |
|---|---|---|---|
| Distance determined by | depth of augmentation chain $d$, mixing parameter $\lambda$ | mixing parameter $\gamma$ | max $\ell_\infty$ norm $\epsilon$ |
| Distance buckets | $S_{d, \lceil \lambda N \rceil}$ | $S_{\lceil 2N \cdot (\min(\gamma, 1-\gamma)) \rceil}$ | $S_{\lceil \epsilon \cdot \frac{N}{\epsilon_{max}} \rceil}$ |

$\mathcal{D}_V, s \sim \mathcal{U}(S_n)\}$. This validation data $\mathcal{Q}(S_n)$ is used to learn a smoothed label (for the true class) $\tilde{y}_{k=y}(S_n)$ that will be used for any training example transformed by a distance $s \in S_n$. After each training epoch $t$, `AutoLabel` updates $\tilde{y}_{k=y}(S_n)$ according to:

$$\tilde{y}_{k=y}^{t+1}(S_n) = \tilde{y}_{k=y}^t(S_n) - \alpha \cdot \text{ECE}^t(\mathcal{Q}(S_n)) \cdot \text{sign}(\text{Conf}^t(\mathcal{Q}(S_n)) - \text{Acc}^t(\mathcal{Q}(S_n))) \qquad (1)$$

where $\text{ECE}(\mathcal{Q}(S_n))$ is the expected calibration error on the augmented validation set, and $\text{Acc}(Q(S_n))$ and $\text{Conf}(\mathcal{Q}(S_n))$ are the accuracy and confidence on the augmented validation set, respectively. The sign of $(\text{Conf}(\mathcal{Q}) - \text{Acc}(\mathcal{Q}))$ indicates if the model is overall over-confident $(> 0)$ or under-confident $(< 0)$. Intuitively, if the model is overconfident on the validation set, we should reduce the label given to the true class $\tilde{y}_{k=y}$, otherwise we should increase $\tilde{y}_{k=y}$. The expected calibration error on the augmented validation set $\text{ECE}(\mathcal{Q}) \geq 0$ suggests to what extent we should adjust the labels as the optimal result is $\text{ECE}(\mathcal{Q}) = 0$ when the training converges. The hyperparameter $\alpha$ controls the step size of updating the labels. Since $\tilde{y}_{k=y}^{t+1}$ stands for the probability of the true class, we clip the value to be within $[\text{Acc}^t(\mathcal{Q}), 1]$ after each update. $\text{Acc}^t(\mathcal{Q})$ is used as the minimum clipping value to prevent $\tilde{y}_{k=y}^{t+1}$ from being too small as $\text{Acc}^t(\mathcal{Q}) \to \frac{1}{K}$ when the classifier is a random guesser.

Given the updated label for the true class $\tilde{y}_{k=y}^{t+1}$, the labels for other classes are uniformly distributed:

$$\tilde{y}_{k \neq y}^{t+1} = (1 - \tilde{y}_{k=y}^{t+1}) \cdot \frac{1}{K-1}, \qquad (2)$$

where $K$ is the number of classes in the dataset and $\sum_{k=1}^K \tilde{y}_k = 1$. Finally, `AutoLabel` trains the model using $\tilde{y}(S_n)$ as the target for the cross-entropy loss across the augmented data. For each example in the validation dataset $\mathcal{D}_V$, we will sample a distance $s$ in order to generate it's transformed version; we describe below this sampling distribution for each method.

`AutoLabel` can easily slot into existing data augmentation methods and further improve model's generalization and robustness. In the following sections, we demonstrate how to apply `AutoLabel` to learn more appropriate labels over *three representative* data augmentation methods:

- AugMix (Hendrycks et al., 2020): Mixing diverse simple augmentations in convex combinations.
- mixup (Zhang et al., 2018): A linear interpolation of random clean images.
- Adversarial Training (Madry et al., 2017): A special case of data augmentation to improve adversarial robustness via training on constructed adversarial examples.

Table 1 shows an overview of how `AutoLabel` differentiates the augmented data based on the transformation distance under each data augmentation method. We present a complete pseudocode for `AutoLabel` in Algorithm 1 in Appendix.

### 4.2 AutoLabel for AugMix

**An overview of AugMix** AugMix (Hendrycks et al., 2020) augments the input data via feeding the input $x$ into an augmentation chain[1] which consists of $d \in \{1, 2, 3\}$ augmentations randomly sampled from a set of diverse augmentation operations, e.g., color, sheer, translation. Then a convex combination is performed to mix the augmented image $x_{\text{aug}}$ with the original image $x$:

$$\text{Aug}_{augmix}(x) = \lambda \cdot x + (1 - \lambda) \cdot x_{\text{aug}} \qquad (3)$$

where the mixing parameter $\lambda \in [0, 1]$ is randomly sampled from a uniform distribution. The model is trained on the augmented data pairs $\{(\text{Aug}_{augmix}(x), \hat{y})\}$, where $\hat{y}$ is the one-hot encoding of the label associated with the original image.

---

[1]The original AugMix implementation (Hendrycks et al., 2020) uses 3 augmentation chains and then mixes the results from each of the augmentation chain via convex combinations. However, we consistently observe an accuracy increase when we use one augmentation chain. Hence, we use AugMix with one augmentation chain.

**Transformation Distance** In AugMix, the transformation distance is controlled by two parameters: (1) the depth of the augmentation chain $d$, which decides how many augmentation operations are applied to the original image; (2) the mixing parameter $\lambda$ in Eqn (3), which controls the ratio of the augmented image $x_{\text{aug}}$ and the original image $x$. In Figure 1(b) (ii) and (iv), we show the image that augmented by a augmentation chain with depth $d = 1$ and $d = 3$ without mixing with the original image ($\lambda = 0$). We can clearly see that a deeper augmentation chain causes the image to quickly degrade and drift off the data manifold. In addition, when comparing the augmented images with different mixing parameter $\lambda$, shown in Figure 1(b) (iii) and (iv), we can see that as $\lambda \to 0$, the augmented image is further away from the clean image. As a result, we can define the distance bucket $S_{d,n}$ for the augmented data as: $S_{d,n} = S_{d,\lceil \lambda N \rceil}$.[2] Note here $S$ is indexed by both $d$ and $\lambda$ since the distance relies on both the depth of the augmentation chain $d$ and the mixing parameter $\lambda$, and in total we have $N \cdot d_{max}$ buckets, where $d_{max}$ denotes the maximum depth of the augmentation chain.

**Update Labels** To learn the labels for augmented training data that are within a distance bucket $S_{d,n}$ to the clean distribution, `AutoLabel` constructs an augmented validation set $\mathcal{Q}(S_{d,n})$ by feeding the validation images into an augmentation chain with the depth $d$ and then randomly sample a mixing parameter $\lambda'$ from a uniformly distribution: $\lambda' \sim \mathcal{U}(\frac{n}{N}, \frac{n+1}{N})$ to mix the original image and the augmented image as Eqn (3). Finally, `AutoLabel` updates the labels $\tilde{y}(S_{d,n})$ according to Eqn (1) & (2) and train the model using these updated labels.

### 4.3 AUTOLABEL FOR MIXUP

**Overview of mixup** Mixup, originally proposed by Zhang et al. (2018), can consistently improve classification accuracy and further has been shown to be able to help with calibration in (Thulasidasan et al., 2019). Specifically, the input data as well as their labels are augmented by

$$
\begin{aligned}
\text{Aug}_{mixup}(x_i, x_j) &= \gamma x_i + (1 - \gamma)x_j \\
\text{Aug}_{mixup}(y_i, y_j) &= \gamma \hat{y}_i + (1 - \gamma)\hat{y}_j
\end{aligned}
\tag{4}
$$

where $x_i$ and $x_j$ are two randomly sampled input data and $\hat{y}_i$ and $\hat{y}_j$ are their associated one-hot labels. The model is trained with the standard cross-entropy loss $\mathcal{L}(f(x_{mixup}), y_{mixup})$ and the mixing parameter $\gamma \in [0, 1]$ that determines the mixing ratio is randomly sampled from a Beta distribution Beta$(\beta, \beta)$ at each training iteration. Rather than using the same mixing parameter $\gamma$ to combine the labels, we show how to apply `AutoLabel` to automatically learn its labels $\text{Aug}_{mixup}(y_i, y_j)$ based on the validation calibration.

**Transformation Distance** The transformation distance in mixup is determined by the mixing parameter $\gamma$ in Eqn (4). When $\gamma \to 0.5$, combining two images equally, the augmented image $\text{Aug}_{mixup}(x_i, x_j)$ is the most far away from the clean distribution. On the other hand, when $\gamma \to 0$, the augmented image is close to the original image. Hence, the distance bucket $S_n$ for each augmented example $\text{Aug}_{mixup}(x_i, x_j)$ can be defined as: $S_n = S_{\lceil 2N \cdot (\min(\gamma, 1-\gamma)) \rceil}$.

**Update Labels** To learn the labels for the augmented training image within the distance bucket $S_n$, `AutoLabel` constructs an augmented validation set $\mathcal{Q}(S_n)$ by randomly mixing two images from validation data with a mixing parameter $\gamma'$ that is sampled from a uniform distribution: $\gamma' \sim \mathcal{U}\left(\frac{n}{2N}, \frac{n+1}{2N}\right)$ and $\gamma' \in [0, 0.5]$. Unlike AugMix, there are two classes $y_i$ and $y_j$ existing in the augmented image $\text{Aug}_{mixup}(x_i, x_j)$. Due to $\gamma' \in [0, 0.5]$, the class in the image $x_j$ plays a dominant role in determining the main class in $\text{Aug}_{mixup}(x_i, x_j)$. Therefore, we follow Eqn (1) to update the label $\tilde{y}_{k=y_j}$ for the class $k = y_j$. Unlike Eqn (2) that uniformly distributes the probability $1 - \tilde{y}_{k=y_j}$ to all other classes, we update the label for the class $k = y_i$ as $\tilde{y}_{k=y_i} = \min(1 - \tilde{y}_{k=y_j}, \frac{\gamma'}{1-\gamma'}\tilde{y}_{k=y_j})$ and then distribute the probability $1 - \tilde{y}_{k=y_i} - \tilde{y}_{k=y_j}$ to all other $K - 2$ classes. Finally, the model is trained by minimizing the cross-entropy loss with the new labels $\tilde{y}$ as the target.

### 4.4 AUTOLABEL FOR ADVERSARIAL TRAINING

**Overview of Adversarial Training** Adversarial training (Goodfellow et al., 2014) can be formulated as solving the min-max problem: $\min_w \mathbb{E}_{||\delta||_\infty \leq \epsilon}[\max \mathcal{L}(f(x + \delta; w), y)]$, where $\delta$ denotes the adversarial perturbation, $\epsilon$ denotes the maximum $\ell_\infty$ norm and a one-hot encoding of the label $y$

---

[2]In the special case where $\lambda = 0$, we merge it into bucket $S_1$ to avoid creating an additional bucket which only contains data under this special case. Similar operation applied to $\gamma$ in mixup and $\epsilon$ in adversarial training.

is used as the target for the cross-entropy loss $\mathcal{L}$. In (Madry et al., 2017), the inner maximization problem is approximately solved by generating projected gradient descent (PGD) attacks. Therefore, adversarial training can be considered as a specific data augmentation that aims for improving model's adversarial robustness.

**Transformation Distance**   First, `AutoLabel` differentiates the adversarial examples according to the distance between the adversarial examples and clean data. In adversarial training, the distance is approximately captured by the $\ell_\infty$ norm of the adversarial perturbation $\epsilon$. Unlike Madry et al. (2017) using a fixed $\epsilon$ to construct PGD adversarial attacks during training, we randomly sample $\epsilon$ from a uniform distribution $\epsilon \sim \mathcal{U}(0, \epsilon_{max})$ to construct PGD adversarial attacks with different distances to the original data. If the $\ell_\infty$ norm of the adversarial perturbation is bounded by $\epsilon$, then for those constructed adversarial examples, we can determine their distance bucket $S_n$ as: $S_n = S_{\lceil \epsilon \cdot \frac{N}{\epsilon_{max}} \rceil}$.

**Update Labels**   To learn the labels for adversarial examples within a distance bucket $S_n$ to the clean data, `AutoLabel` constructs adversarial examples for the validation set with the $\ell_\infty$ norm of the adversarial perturbation bounded by $\epsilon'$, where $\epsilon'$ is randomly sampled from a uniform distribution: $\epsilon' \sim \mathcal{U}(\frac{n \cdot \epsilon_{max}}{N}, \frac{(n+1) \cdot \epsilon_{max}}{N})$. Then we can update the training labels $\tilde{y}(S_n)$ for adversarial examples following Eqn (1) and (2) and train the model using cross entropy loss on these new labels.

## 5   EXPERIMENTS

### 5.1   DATASETS AND EVALUATION METRICS

**Datasets**   We perform experiments over three widely-used datasets: CIFAR10 (Krizhevsky, 2009), CIFAR100 (Krizhevsky, 2009), and ImageNet (Russakovsky et al., 2015). We use a Wide ResNet-28-10 v2 (Zagoruyko & Komodakis, 2016) for both CIFAR datasets, and a ResNet-50 (He et al., 2016) for ImageNet as our basic model architectures. In addition, we test our model's robustness over a set of corrupted datasets (Hendrycks & Dietterich, 2019) with different types (17 types for CIFAR10 and CIFAR100 and 15 types for ImageNet) of corruptions that are frequently encountered in natural images. Each type of corruption has five levels of corruption severities.

**Evaluation Metrics**   We report the classification accuracy and expected calibration error on the clean datasets as well as the corrupted datasets, denoted as **Acc** (higher is better) and **ECE** (lower is better) respectively. The accuracy and expected calibration error on the corrupted datasets are represented as **cAcc** (higher is better) and **cECE** (lower is better), which are computed as an average over all the corruption types across 5 corruption severities in the corrupted datasets. When evaluating adversarial robustness, we report the accuracy of the model against white-box PGD attacks (Madry et al., 2017). These attacks are generated by 50 iterations and 3 random restarts bounded by $\epsilon \in \{0.02, 0.03, 0.04\}$ with the image scale is $[0, 1]$. The step size is $\epsilon/4$.

### 5.2   AUTOLABEL IMPROVES THE CALIBRATION OF DATA AUGMENTATION TECHNIQUES

In this section, we apply `AutoLabel` to three data augmentations: AugMix, mixup and adversarial training, and show that `AutoLabel` can help improve model's generalization and robustness.

**Improve accuracy and calibration on clean data**   We first investigate if `AutoLabel` can make data augmentation approaches more effective in improving accuracy and calibration on clean test data. For AugMix, we see in Table 2 a clear picture: `AutoLabel` consistently helps AugMix improve both accuracy and calibration across CIFAR10, CIFAR100 and ImageNet.

Table 2: Effects of `AutoLabel` for AugMix on CIFAR10, CIFAR100 and ImageNet and the corresponding corrupted datasets. All numbers reported in the table is an average of 4 independent runs and in %. The arrow indicates better direction. Best results are highlighted in **Bold**.

| Method | Acc/cAcc (↑) | | | ECE/cECE (↓) | | |
|---|---|---|---|---|---|---|
| | CIFAR10 | CIFAR100 | ImageNet | CIFAR10 | CIFAR100 | ImageNet |
| **AugMix** | 96.9/87.7 | 80.6/63.9 | 76.3 /47.0 | 1.0/4.1 | 5.1/11.8 | 1.9/5.5 |
| **+ AutoLabel** | **96.9/88.2** | **81.6/65.0** | **76.7/47.6** | **0.9/2.7** | **1.8/4.3** | **1.4/4.3** |

To test how well `AutoLabel` improves mixup (Zhang et al., 2018) is more nuanced because mixup's baseline effectiveness is sensitive to its hyperparameters. In particular, the mixing parameter $\gamma$ in Eqn (4) is sampled from a beta distribution Beta$(\beta, \beta)$. When $\beta \to 0$, most sampled $\gamma$ are close to 0

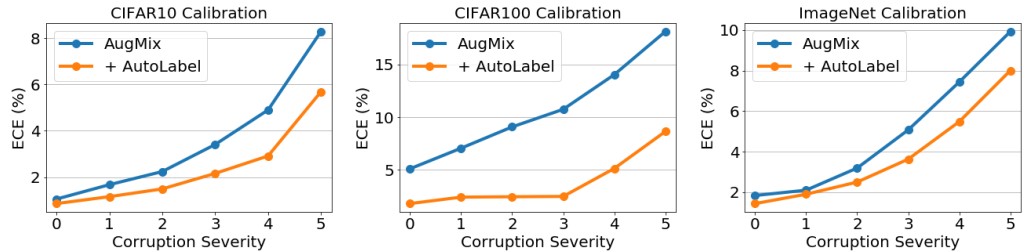

Figure 2: Calibration across corruption severities on three datasets. Severity 0 denotes clean data. The improvment of `AutoLabel` over AugMix is more significant as corruption severity increases.

Table 3: Effects of `AutoLabel` for mixup on CIFAR10 and CIFAR100. All numbers reported are averaged over 4 independent runs and in %. Best highlighted in **bold**.

| Method | CIFAR10 | | CIFAR100 | |
|---|---|---|---|---|
| | Acc | ECE | Acc | ECE |
| **Vanilla** | 95.6 | 2.6 | 79.5 | 6.1 |
| **mixup** ($\beta = 0.2$) | 96.2 | 0.8 | 80.8 | 1.8 |
| **mixup** ($\beta = 1$) | 96.5 | 5.3 | 80.9 | 5.5 |
| **+ AutoLabel** ($\beta = 1$) | **96.7** | **0.6** | **81.1** | **1.2** |

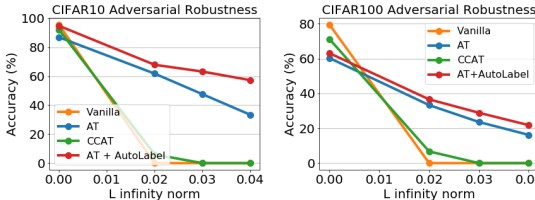

Figure 3: Effects of `AutoLabel` for adversarial training on CIFAR10 and CIFAR100. Each model is tested on the $\ell_\infty$ norm based white-box PGD attacks generated by 50 iterations and 3 random restarts. The $x$ axis is the maximum $\ell_\infty$ norm. 0 denotes the clean data.

or 1; when $\beta = 1$, $\gamma$ is randomly sampled from a uniform distribution. We observe that mixup suffers from a trade-off between accuracy and calibration on the clean data on CIFAR10 and CIFAR100, shown in Table 3. When the hyperparameter $\beta$ is large, e.g., $\beta = 1$, the model is trained on more diverse augmented data compared to a smaller $\beta$, e.g., $\beta = 0.2$. This results in a higher accuracy but leads the model to be too under-confident and a much larger calibration error on the clean data. This trade-off between clean accuracy and calibration of mixup is also observed in other datasets and networks in Figure 2(j) in Thulasidasan et al. (2019). After applying `AutoLabel` to mixup to automatically adjust the labels for augmented images, we find that the trade-off between accuracy and calibration is well addressed: high accuracy and low calibration error are achieved on the clean data, shown in Table 3. However, this trade-off between accuracy and calibration does not exist in the large scale datasets, e.g., ImageNet, where $\beta = 0.2$ consistently has the best accuracy and calibration and we did not observe a significant improvement when applying `AutoLabel` to mixup on ImageNet.

**Improve robustness under distributional shifts** While top-line improvements on accuracy are valuable, we'd like to also understand how this method is getting these improvements, and one key way is to see the benefits on corrupted data that the approach is more directly targeting. When applying `AutoLabel` to AugMix, we again see a very clear picture in Table 2: on three corrupted datasets, `AutoLabel` significantly improves both accuracy and calibration under distribution shifts. We similarly find benefits when applying `AutoLabel` to mixup: applying `AutoLabel` to mixup with $\beta = 1$, we can further improve the mean accuracy on CIFAR100-C over mixup with $\beta = 0.2$ from 55.6% to 56.9% and reduce the mean calibration error from 11.2% to 10.0%. Similar results on CIFAR10-C are shown in Table 2 in Appendix.

Digging into these results further, we find evidence of why `AutoLabel` is working. In Figure 2 we analyze how calibration performance changes with the severity of the corruption being tested against, comparing AugMix and AugMix + `AutoLabel`. We see that the baseline AugMix approach is increasingly badly calibrated as the corruption increases, but `AutoLabel`, as intended, dampens that trend significantly, and is thus increasingly important as the corruption size increases.

**Improve trade-off between adversarial robustness & accuracy** Similar to human-designed data augmentation, we also believe adversarial training runs the risk of attacks not being truly label-preserving (Qin et al., 2020a) and thus should benefit from setting labels carefully. Therefore, we would like to see if `AutoLabel` can improve the trade-off between adversarial robustness and accuracy. As a baseline, we perform adversarial training with PGD attacks as (Madry et al., 2017) on CIFAR10 and CIFAR100 datasets. Then we test the models against white-box untargeted PGD attacks bounded with different $\ell_\infty$ norms. We compare adversarial training (AT) with and without `AutoLabel` in Figure 3. There we see that `AutoLabel` effectively helps improve the trade-off

Table 4: Effects of `AutoLabel` for adversarial training under distributional shifts. All the numbers reported are an average over 2 independents runs and in %. The best result is highlighted in **bold**.

| Method | Acc/cAcc($\uparrow$) | | ECE/cECE($\downarrow$) | |
|---|---|---|---|---|
| | CIFAR10 | CIFAR100 | CIFAR10 | CIFAR100 |
| **Vanilla** | **95.6**/76.0 | **79.5**/52.0 | 2.6/15.8 | 6.1/17.6 |
| **AT** | 93.6/**83.9** | 71.5/58.1 | 3.7/10.5 | 8.0/13.5 |
| **CCAT** | 93.2/68.9 | 74.8/49.8 | 2.4/9.9 | 7.9/16.1 |
| **AT + `AutoLabel`** | 94.6/83.6 | 75.3/**60.2** | **2.0/6.5** | **4.2/6.9** |

between adversarial robustness against PGD adversarial attacks and accuracy, achieving higher accuracy on both clean and different levels of adversarial attacks.

We also compare our method with the recent work confidence-calibrated adversarial training (CCAT) (Stutz et al., 2020), which smooths the labels for adversarial examples according to $\tilde{y} = g(\delta)\hat{y} + (1 - g(\delta)\frac{1}{K})$, where the balancing parameter $g(\delta)$ follows a "power transition": $g(\delta) := (1 - \min(1, \frac{||\delta||_\infty}{\epsilon}))^\rho$, $\rho$ is set to 10 as (Stutz et al., 2020) to ensure a uniform distribution is used as the labels for adversarial examples when $||\delta||_\infty \geq \epsilon$. CCAT is shown to help improve adversarial accuracy after confidence-thresholding. However, using the uniform distribution for adversarial examples results in a loss of class information and leads to a poor adversarial robustness without any confidence-thresholding detection mechanism, as shown in Figure 3.

**Improve adversarial training to be beneficial to calibration**   We further validate if adversarial training can improve model's robustness under distribution shifts. Similar to recent work (Xie et al., 2020), we observe that training models with adversarial examples bounded with smaller $\ell_\infty$ norm, e.g., $||\delta||_\infty \leq 0.01$, can benefit more to the corrupted accuracy with a small accuracy sacrifice on the clean data. Therefore, we train all models with PGD attacks bounded by $||\delta||_\infty \leq 0.01$ updated with 10 iterations. In Table 4, we present the results of four models: a vanilla model without adversarial training, adversarial training (AT) (Madry et al., 2017), confidence-calibrated adversarial training (CCAT) (Stutz et al., 2020) and adversarial training + `AutoLabel`. When comparing the accuracy on the clean and corrupted datasets, `AutoLabel` consistently has a better balance of classification accuracy between clean and corrupted data. In addition, `AutoLabel` has a significant improvement over model's calibration performance with a much smaller calibration error on both clean and corrupted data compared to vanilla model, AT and CCAT. We explains this as a result of that `AutoLabel` updates training labels based on the validation calibration performance. In constrast, a pre-defined function to smooth the labels, e.g, the power transition used in CCAT, does not significantly help model's calibration.

An extensive ablation study comparing `AutoLabel` with label smoothing (Szegedy et al., 2016; Müller et al., 2019) as well as temperature scaling (Guo et al., 2017) are included in Section B in Appendix, where we find that `AutoLabel` consistently has a better improvement over model's calibration on both clean and corrupted datasets. Note that temperature scaling, as a post-hoc calibration method, does not help improve model's accuracy, where `AutoLabel` significantly helps.

Taken together, we believe this is strong evidence that `AutoLabel` can improve the effectiveness of data augmentation techniques to improve models' robustness under distribution shifts, especially calibration performance.

## 6 CONCLUSION

In this paper, we propose `AutoLabel` to automatically adapt labels for augmented data, and show it is beneficial to both model accuracy and calibration, compared to reusing one-hot labels as in common in existing data augmentation works. We found that the distance between the augmented data and the clean data is key to how we should adapt the labels and to what extent, and we propose a calibration-guided approach to further adjust the labels automatically. We demonstrate the effectiveness of `AutoLabel` under three representative data augmentation methods: AugMix, mixup, and adversarial training. We see that `AutoLabel` greatly improves the calibration over clean data and corrupted data, and also helps adversarial training with a better trade-off between clean accuracy and adversarial robustness. More generally, we believe that more nuanced approaches to setting labels for augmented data, beyond assuming label-preserving transformations, will lead to more effective data augmentation techniques, and look forward to further work in this space.

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
