# OpenReview forum: "What are effective labels for augmented data? Improving robustness with AutoLabel"
_ICLR.cc/2021/Conference — Reject_

### Official Review · AnonReviewer2 · 2020-10-27
**Weak due to the insufficient experimental studies and the minor performance improvement obtained over baseline models**

**Rating:** 4
**Confidence:** 4

**Review:**

The paper proposed a new method, called AutoLabel, to assign labels to synthetic samples. The label assignment is calculated by the distance between the clean data and the augmented data, and updated by performing a label smoothing based on the calibration performance over a hold-out-validation data set. Experiments show performance improvement over vanilla baseline model.

The paper is well motivated and the problem is important to the community. But the current form of the paper is weak due to the insufficient experimental studies and the minor performance improvement obtained with additional computational cost, compared to the baseline models. I have the following major comments.

1.	The proposed label assignment is closely related to label smoothing as emphasized in the related work section and the main text (“AutoLabel takes a label smoothing approach where the amount of smoothing …”). I think it would be useful to use label smoothing as a main comparison baseline in the experiments. I saw some experiments with Cifar10 and Cifar100 in the Supplementary Material, but I think they should be moved to the main text.  Also, comparison with the label smoothing on Cifar10 show very similar results (but the label smoothing is much easier to be implemented), so it would useful to also provide the comparison result on ImageNet. Also, I think the idea is a bit similar to the Structural label smoothing technique (Reguralization via structural label smoothing, Li et al., 2020), where the data is first clustered and then the approach assigns different smoothing mass to different clusters formed. I think this may also be a valid comparison baseline as well.

2.	One main disadvantage of the proposed method is that it seems to be sensitive to the distance function chosen for the label assignment and this function has to be adjusted for different data augmentation schema. The current justification for the three data augmentation schemas namely AugMix, MixUp, and Adversarial Training is weak to me. For example, in Mixup, the random mixing ratio may not reflect the salient features of the mixed image and leverage this ratio for the distance function for label assignment could be misleading sometimes (see “Mixup as locally linear out-of-manifold regularization”, AAAI2019, for more detail). In this sense, in the experiments, it would be very useful to provide further analysis and discussion of these distance functions in terms of how they impact the model’s performance. Also, I wonder if it makes sense to leverage the saliency in the mixed samples to compute the distance function. Leveraging salient features have been shown to be promising by recent research such as in the Puzzle Mix method (ICML 2020).

3.	The way the paper generates the mixed modeling target for the mixed input in Mixup is related to that proposed in this paper: nonlinear mixup: out-of-manifold data augmentation for text classification,  AAAI2020. In the nonlinear Mixup method, the mixed label for a mixed input in Mixup is assigned by computing a transformation based on the inputs. It would be beneficial to briefly discuss the difference of the two methods for label assignment in Mixup.

4.	The experiments are a bit weak to me. I think experiments with some additional network architectures, such as ResNet-18 in Table 2, would improve the paper.

---

> ### Author Response · Authors · 2020-11-24
> **To Reviewer2**
>
> We want to emphasize that AutoLabel is a generic algorithm which can be easily applied to existing data augmentation methods. The major benefit of AutoLabel is its strong generalization ability across data augmentations. This is supported by the experimental results that AutoLabel can improve the robustness of three different data augmentation techniques. Mixup is one of the data augmentation techniques but the effectiveness of AutoLabel is not limited to mixup. In all, we want to focus more on the generalization of the algorithm.

---

### Official Review · AnonReviewer3 · 2020-10-28
**Official Blind Review #3**

**Rating:** 5
**Confidence:** 4

**Review:**

This paper proposes a method to adaptively smooth the augmentation data labels based on the augmentation strength. The motivation is that the label-preserving assumption may not hold for data with substantial augmentations. The labeling smoothing uses the calibration performance on the validation data. If the model is overconfident on the validation data, the method will enforce more considerable smoothing. The paper applies it to three popular data augmentation methods: AugMix, Mixup, and adversarial training. Experiments on CIFAR-10, CIFAR-100, and ImageNet show the improvements for both accuracy and calibration.

Pros
1. Automatically tuning the labels for augmented data is interesting, as most previous works mainly focus on new augmentation methods.
2. The proposed method is general, applicable to different types of augmentations: AugMix, Mixup, and adversarial training.
3. Experiments demonstrate its effectiveness on benchmark image classification datasets measured by accuracy and calibration.

Cons
1. Although divided into N buckets, estimating the confidence and accuracy for a transformation bucket may still need several samples. The paper does not discuss this critical factor and provides an ablation study on it.
2. It needs more justification to use validation data's calibration for label smoothing. What's its advantage compared with other tools?
3. Why does it require 2N divisions for the Mixup design? I don't see a clear link between 2 samples in Mixup and using 2N here.
4. More explanation on the equation to update the label y_{k=y_i} in the Mixup part is necessary.
5. According to the experiments, the accuracy improvements are small. Calibration errors drop clearly because the validation calibration information is used in training. I don't see the description of the train-validation split.
6. Ablation of some key hyper-parameters are missing, e.g., the bucket number N. The experiment section seems not to mention the chosen value of N.
7. The label smoothing rule, i.e., Equation 1, is similar to previous work 1, which smoothes labels for adversarial examples. Technically, this paper mainly applies the idea to more augmentation settings.

Summary

Challenging the label-preserving assumption differentiates the proposed method from a majority of works in the data augmentation area. The proposed method is general to adjust labels for different data augmentation methods. My concerns lie in the technical novelty and lack of clarifications on the method design and experimental setup. Also, the empirical performance on accuracy does not show apparent improvements. More explanations on this may be better. Overall, I currently hold my score as marginally below the acceptance threshold.


Reference

[1]. Improving uncertainty estimates through the relationship with adversarial robustness. Arxiv 2020.

---

### Official Review · AnonReviewer1 · 2020-10-28
**A straightforward and simple method to determine the label for augmented samples, but sometimes unclear motivation and limited novelty. Experiments for adversarial training are not performed in standard setting.**

**Rating:** 4
**Confidence:** 3

**Review:**

Summary:
This paper proposes to use label smoothing to determine the labels of augmented samples. The distance between the original data distribution and the augmented data distribution is used to determine the amount of smoothing in the smoothed label. This technique can be used in combination with several data augmentation techniques e.g. AugMix, Mixup, and adversarial training.

Strength:
- This paper is structured clearly and easy to follow.
- The idea is straightforward and the proposed distance measures for the three data augmentation methods are straightforward.

Weakness:
- The smoothed label in equation (1) and (2) of each augmented sample is based on an augmented validation set $Q(S_{n})$ which however directly uses the validations samples' original labels. Is this a contradiction to the main motivation of using a smoothed label for augmented data?

- Using smoothed label for augmented data is not quite novel,  and has already being used in combination with adversarial training in [1]. Can the authors please clarify the connection and difference with this paper?

- The accuracy results of corrupted data (adversarial samples) in Table 4 are not very convincing. The  $l_{\infty}$ perturbations are upper bounded by only 0.01, while the commonly used perturbation bound is 8/255. Thus the attack in this paper may not strong enough, thus making the robustness conclusion not convincing. Can the authors show results using a larger commonly used perturbation bound?

-  We can avoid accessing the label of adversarial samples for adversarial corrupted data. For example, TRADES[2] substitutes the original label with the model's output on clean data when training with corrupted data. Hence the proposed method may not be very useful for adversarial training.

Reference:
- [1]. Cheng, Minhao, et al. "Cat: Customized adversarial training for improved robustness." arXiv preprint arXiv:2002.06789, 2020.

- [2]. Zhang, Hongyang, et al. "Theoretically Principled Trade-off between Robustness and Accuracy." ICML. 2019.

---

> ### Author Response · Authors · 2020-11-24
> **To Reviewer1**
>
> First, we want to point out that AutoLabel is a very generic framework that can be combined with the existing data augmentation methods. Adversarial training is one of these data augmentation methods that AutoLabel can help but it is not the only one.
>
> Second, we want to point out that we do follow the standard setting in adversarial literature by using 8/255 (~0.03) as the l infinity bound of the adversarial perturbation and the result is shown in Figure 3. In addition, Table 4 is not to show the performance of adversarial robustness but to show AutoLabel can “Improve adversarial training to be beneficial to calibration”, highlighted in page 8. We also provide a explanation as “Similar to recent work (Xie et al., 2020), we observe that training models with adversarial examples bounded with smaller l∞ norm, e.g., ||δ||∞ ≤ 0.01, can benefit more to the corrupted accuracy with a small accuracy sacrifice on the clean data. ”
>
> We answer your questions as follow:
> Is using the original labels of the validation set a contradiction to the main motivation of using a smoothed label for augmented data? No, there is not any contradiction. First, AutoLabel builds on the observation that one-hot vector is not an appropriate choice to be used as the labels for highly corrupted augmented data during training because one-hot vector assumes the confidence in the label to be 100%. Next, we will update the training labels based on the calibration performance on the augmented validation set. In this stage, we only need the semantic class of the validation set rather than the one-hot labels to compute the accuracy and ECE.
>
> Novelty: label smoothing is a well-known method even in adversarial training. The major differences between Cheng et al and ours are 1) our method is a generic framework that can be applied to many different data augmentation methods, not limited to adversarial training. 2) AutoLabel adjusts the labels based on the calibration performance on the augmented validation set, which can significantly improve models’ calibration under distributional shifts. However, Cheng et al only focuses on improving adversarial robustness by adjusting the labels with a simple linear function.
>
> Standard Setting: The result showing AutoLabel can help improve adversarial robustness is originally provided in Figure 3.
>
> Not effective for adversarial training: We have shown that AutoLabel can improve the trade-off between adversarial robustness & accuracy by applying AutoLabel to standard adversarial training which uses one-hot labels during training. In addition, standard adversarial training can achieve comparable adversarial robustness with early stopping compared to TRADES in [1]. Therefore, we argue that AutoLabel is useful for adversarial training.
>
> [1] Rice, Leslie, Eric Wong, and J. Zico Kolter. "Overfitting in adversarially robust deep learning." ICML (2020).

---

### Official Review · AnonReviewer4 · 2020-10-28
**Strong motivation with strong lack of clarity and support of the technique**

**Rating:** 4
**Confidence:** 4

**Review:**

Review:

[Summary] In this work, authors address the problem of how to achieve a more feasible data augmentation not only from the samples but also considering the accompanying labels. Authors solution is framed on a framework called ‘AutoLabel’, in which they leverage on already existing concept including: mainly on expected calibration error, and AugMix (in general mixup principles).

[Cons]
-- The problem of adjusting labels for augmented data is indeed of interest for the community.

-- The motivation of the work is good.


[Pros]

-- Several strong statements are not supported, and therefore, the level of novelty is hard to appreciate.

-- There is a lack of formalism to transmit the idea whilst there is empirical evidence of a few statements- there is not enough theoretical background to support the authors findings.



Detailed comments for authors:

-- [Novelty of the work] The proposed method strongly leverages the widely used  expected calibration error along with principles of mixup. Therefore, it is hard to appreciate the significance of the proposed method. Moreover, central observations are unsupported:
 > A major observation of the authors is: “Our key insight is that the confidence in the labels associated with the augmented data likely depends on how significant the transformation is.”  This is the definitional concept behind when generating pseudo-labels (-examples). That is, one seeks to get a high confidence (high probability to belong to the k class) to produce usable augmentation -- even from a set of feasible augmentation one can further average on the confidence produced on the label.  Authors should further elaborate on this idea as this is reported as highlighted in the paper.
> The author's statement “assumption of label-preservation holds in practice” this also echoes the previous point, a major consideration is a ‘feasible augmentation’. This is a major area not only in the supervised setting  but also in semi-supervised learning -- how to define these $\delta$ augmentations (perturbations) is far from being trivial not only in the sense of the augmented labels but also samples. Authors should formalise the assumption in the work including a ‘feasible augmentation’ of the samples (not the labels) as it is highly dependent on the augmented label; strong and weak augmentations how these concepts hold in the paper?
> Author central contribution is given in (1) and the following sections revise concepts and connections with the principles of mixup. Authors should focus on giving a further interpretation of (1) along with the  assumptions that hold and how a certain modelling hypothesis fails this expression.

-- What are the connections with SOTA augmenters? For example RandAugment, and CTAugment.

[*] Cubuk, Ekin D., et al. "Randaugment: Practical automated data augmentation with a reduced search space." CVPRW. 2020.

-- From [**] the randomisation test where the true labels were replaced by random labels yet deep nets easily fits them raised the question of -- Are augmented labels really  better than search for feasible augmented samples (instead of labels)?  If one defines a feasible augmented sample then the label preserving will hold.

[**] Zhang, Chiyuan, et al. "Understanding deep learning requires rethinking generalization." ICLR 2017.

[Experimental Results]  There is a lack of strong discussion and findings that somehow weakens the paper. One would like to see connections with other augmenters such as RandAugment, CTAugment, Cutout etc.

---

> ### Author Response · Authors · 2020-11-24
> **To Reviewer 4**
>
> Connections with RandAugment and CTAugment: We show AutoLabel can significantly improve the performance of Augmix, which includes the same types of transformations included in RandAugment and CTAugment, while Augmix is the most state-of-the-art data augmentation for improving model’s robustness.
>
> Feasible augmented samples then the label preserving will hold: We agree with the opinion that if we augment the training data within a limited transformation distance, then the label preserving will more often hold. However, we want to point out that those highly corrupted augmented training data are critically necessary for improving a model’s robustness on the highly shifted test data. Therefore, simply removing those highly corrupted augmented training is not the best choice.

---

### Author Response · Authors · 2020-11-24
**General Response**

We propose AutoLabel as a generic algorithm, which can be easily applied to existing data augmentation methods. This is supported by the experimental results when we apply AutoLabel to three representative data augmentations. We want to emphasize that the major benefit of AutoLabel is that it is a generic framework and can be easily applied to the existing data augmentation.

---

### Decision · Program_Chairs · 2021-01-07
**Final Decision**

**Decision:**

Reject

**Comment:**

The paper introduces a simple and interesting method that adaptively smoothes the labels of augmented data based on a distance to the “clean” training data. The reviewers have raised concerns about limited novelty, minor improvement over baselines, and insufficient experiments. The author’s response was not sufficient to eliminate these concerns. The AC agrees with the reviewers that the paper does not pass the acceptance bar of ICLR.